# Experimental Study on Water Recovery from Flue Gas Using Macroporous Ceramic Membrane

**DOI:** 10.3390/ma13030804

**Published:** 2020-02-10

**Authors:** Chao Cheng, Heng Zhang, Haiping Chen

**Affiliations:** School of Energy, Power and Mechanical Engineering, North China Electric Power University, Beijing 102206, China

**Keywords:** ceramic membrane, water recovery, flue gas, heat transfer coefficient

## Abstract

In this work, a ceramic membrane tube with a pore size of 1 μm was used to conduct experimental research on moisture and waste heat recovery from flue gas. The length, inner/outer diameter, and porosity were 800 mm, 8/12 mm, and 27.2%, respectively. In the experiments, the flue gas, which was artificially prepared, flowed on the shell side of membrane module. The water coolant passed through the membrane counter-currently with the gas. The effects of flue gas flow rate, flue gas temperature, water coolant flux, and water coolant temperature on the membrane recovery performance were analyzed. The results indicated that, upon increasing the flue gas flow rate and its temperature, both the amount of recycled water and the recovered heat increased. The amount of recycled water, recycled water rate, recovered heat, and heat recovery rate all decreased as the water coolant temperature increased. When the water coolant temperature exceeded 30 °C, the amount of recycled water dropped sharply. The maximum amounts of recycled water, recovered heat, and total heat transfer coefficient were 2.93 kg/(m^2^·h), 3.63 kW/m^2^, and 224.3 W/(m^2^·K), respectively.

## 1. Introduction

With the population growth and rapid economic development, power demand increased year after year [1]. In the case of serious water pollution and gradual water shortage, the contradiction between water demand and water supply shortage in the power production process is deepening [2]. The installed power capacity (above 6 MWe) in China reached 1.86 billion kilowatts of which thermal power accounted for 62.8%, and thermal power generation contributed to 71.8% of total power generation by the end of September 2019 [3]. Based on China’s energy structure, thermal power retained a dominant role for a long time. The boiler flue gas of thermal power plants usually contains a large amount of vapor and lots of latent heat. The vapor in coal-fired boilers and gas-fired boilers accounts for 4–13% and 15–20%, respectively [4]. The discharge of exhaust gas into the atmosphere not only causes waste of water resources, but also leads to wet plume formation, resulting in visual pollution and chimney corrosion [5].

In order to reduce the water consumption of thermal power generation plants, the cooling system of the thermal plant was reformed. For example, in the southern region, the cooling system was changed from open type to closed type. In the northern region, the cooling system was changed from closed type to air cooling type. However, on the whole, water consumption hardly changed [6]. Therefore, extracting moisture from exhaust gas for reuse is a sensible choice to save water and can greatly reduce the dependence of the thermal power industry on freshwater resources.

The technologies of flue gas water recovery currently include condensation cooling, adsorption, and membrane separation technology. The recovered water based on the condensing cooling technology is acidic. Thus, corrosion-resistant materials such as fluorine plastic [7] and polypropylene coatings [8] are required. At present, the production of adsorbent in the adsorption technology requires a large amount of energy; thus, adsorption technology is less economical. Additionally, the treatment of precipitate formed by the contact between the flue gas and the absorbent is technically difficult [9,10]. Usually, the membrane materials used in flue gas moisture recovery are fiber membranes and ceramic membranes. Although the membranes are relatively expensive to manufacture, they take advantages of high efficiency, reliability, and heat and chemical resistance [11,12]. Therefore, membrane separation technology is more promising in the flue gas moisture recovery field.

Regarding fiber membranes, Sijbesma et al. [13] comparatively studied two membrane bundles composed of polyether block amide (PEBAX^®^ 1074) and sulfonated polyether ether ketone (SPEEK) membrane materials for moisture recovery from power plant exhaust. The results revealed that the performance of SPEEK fiber membrane with a sulfonation degree of 60% was better. Under the real flue gas condition, the vapor removal rate of the membrane was 0.2 to 0.46 L/(m^2^·h). Gao et al. [14] performed an experimental study on a polyether sulfone-sulfonated polyether ether ketone (PES–SPEEK) hollow-fiber membrane applied to recycle water from exhaust gas. The influences on water and waste heat recovery, such as sulfonation degree, coating, filling rate, and length of membrane, were analyzed. The fiber membranes mentioned above are all hydrophilic. For hydrophobic membranes, the Macedonio team constructed a membrane condenser using polyvinylidene fluoride hollow-fiber membranes and carried out simulation calculations [11]. They showed that the water recovery rate could reached 20% when the exhaust temperature drop was less than 5 °C. Brunetti et al. [15] experimentally examined the effect of ΔT (temperature difference between the flue gas and the module) on the water recovery rate. When ΔT ranged from 8 to 15 °C, the water recovery increased from 25% to 60%.

For ceramic membranes, a transport membrane condenser (TMC) was developed for the purpose of water recovery and waste heat utilization, and industrial demonstrations of the TMC were conducted on a gas-fired boiler [16]. Wang et al. [17] carried out a pilot test on a gas-fired boiler with a modified two-stage TMC unit. In the test, simulated flue gas was adopted and consisted of SO_2_, NO_2_, CO_2_, and H_2_O components. The results showed that the vapor recovery rate reached 40–55% and that boiler efficiency was increased by more than 5%. Xiao et al. [18] analyzed the entropy generation of TMC and put forward a calculation model of entropy generation. The results showed that, in most cases, when the heat transfer entropy production was the smallest, the heat transfer performance was the best. Furthermore, the maximum mass transfer entropy production ratio usually corresponded to optimal water recovery performance. Soleimanikutanaei et al. [19,20] proposed a new condensation model and studied the effects of different inlet parameters on heat and mass transfer of TMC numerically.

Furthermore, some researchers experimentally and theoretically studied the TMC by using a single ceramic tube. Wang et al. [21] performed an experimental study on influence factors of inlet air flow and temperature, water coolant flux, and temperature on the property of membrane tube recovery performance. The results presented that the water recovery rate was 20–60% and the waste heat recovery rate was 33–85%. Chen et al. [22] experimentally studied the water recovery performance of ceramic membranes with pore sizes of 20, 30, 50, and 100 nm. It was found that the recovery performance of the 20 nm pore-sized membrane was better. Moreover, Chen et al. [4] researched the condensation heat transfer process of a 20-nm-pore ceramic membrane. Zhou et al. [23] analyzed the heat and mass transfer phenomena in the water recovery process. A mathematical model of mass transfer affecting on heat exchange was established. Instead of carrying out a study using conventional flue gas, Gao et al. [24] carried out a study to research the effect of SO_2_ on ceramic membrane extracting water. A multi-channel ceramic membrane in place of a single membrane was used for experimental studies by Yue et al. [25]. It was revealed that the multi-channel membrane tube had a lower mass transfer rate and heat recovery rate by comparing the two types of tubes. In terms of hydrophilic/hydrophobic treatment on the surface of the ceramic membrane, Hu et al. [26] experimentally demonstrated that the hydrophilic nanoporous ceramic membrane had better condensation heat transfer performance.

The nanoporous ceramic membrane comprises three layers of a substrate, an intermediate layer, and a selective layer. The intermediate and selective layers need to be coated and sintered several times. The sintering temperature is high, and the manufacturing process is complex, leading to a high product cost [27,28,29]. The macroporous ceramic membrane generally consists of a substrate and a thin separation layer, whereby the production cost is relatively low. In order to explore the application value of economical TMC in engineering and reduce the equipment investment cost of flue gas moisture recovery, in this paper, a single-membrane tube with a pore size of 1 μm was used for the experimental study of extracting moisture and waste heat from the exhaust gas. An inner-coating membrane tube was used, i.e., the gas flowed inside the pipe [4,21,22,30]. On the other hand, an outer-coating membrane tube was used here, whereby the flow pattern did change, and water coolant in place of gas flowed inside the tube. In engineering application, the mode in which the flue gas flows inside the tube can make the installation very complicated and increase flue gas flow resistance, which is not suitable for practical application. The effects of flue gas flow rate, flue gas temperature, water coolant flux, and temperature on the ceramic membrane performance were studied. The results of this study can provide guidance for recovering water and heat from power plant exhaust by utilizing ceramic membranes.

## 2. Experiment and Calculation Method

### 2.1. Structural Characterization and Water Recovery Mechanism

Figure 1 presents the scanning electron microscopy (SEM) image of 1-μm-average-pore ceramic membrane. The microstructure (Figure 1) of the membrane tube presented a typical porous structure. The pore size distribution was uniform and the surface was smooth, and there were no cracks and defects.

In the membrane separation technique, for the separation of water vapor, when the membrane aperture is in the range of 0.348 nm to 1 nm, the molecular sieve [31] mechanism plays a leading role. The capillary condensation plays a dominant role when the pore diameter is 2–50 nm [32,33]. When the pore diameter is 50–200 nm, Knudsen diffusion [34] occurs. Furthermore, when the pore size exceeds 1 μm, the water recovery mechanism is surface condensation and permeation. The ceramic membrane tube used herein had an average pore size of 1 μm. Therefore, the process of water recovery was that of condensation on the external tube wall first, before penetrating into the tube along membrane pores.

### 2.2. Experimental System

In this paper, artificial flue gas, including nitrogen and water vapor, was used to investigate the performance of extracting water and heat of membrane from flue gas. The experimental set-up included a membrane module, flue gas section, and water coolant section. Figure 2 illustrates the experimental system schematic.

The membrane module was mainly made up of a ceramic membrane and 316L stainless-steel housing. The ceramic membrane tube was made of α-Al_2_O_3_, the length of the tube was 800 mm, the inner/outer diameter was 8/12 mm, the average pore diameter was 1 μm, the porosity was 27.2%, and the effective membrane area was 0.0294 m^2^.

To prevent heat dissipation, the housing and pipes were covered with an insulation layer. Nitrogen was supplied by a high-purity nitrogen cylinder. It entered the humidifier through the gas flow controller. The humidifier was heated by a thermostatic water bath to maintain flue gas at a constant temperature. After humidification, it flowed into the membrane module through the buffer tank. The temperature and humidity of the gas were measured by the thermo-hygrometers installed at the module entrance. The flue gas was finally discharged to the atmosphere after passing through a silica gel drying bottle.

The water coolant was provided by an insulated water feed tank. The water pump was installed at the outlet side of the water coolant of the membrane tube to maintain a negative pressure inside the tube. The water coolant flowing inside the tube had a counter-current flow against the gas in the shell side. The water coolant was discharged to the return tank. In the experiment in this paper, the relative vacuum was about −20 kPa.

The water coolant temperature in the experiments was less than the flue gas temperature; in addition, the environment inside the membrane tube featured negative pressure. Under the combined action of the temperature difference and pressure difference, water vapor condensed into water, before transiting across the membrane, and then being discharged to the return tank with water coolant. The unrecovered water vapor was assimilated by the silica-gel desiccant. The experimental parameters are shown in Table 1. The measuring devices used in the experiment are listed in Table 2.

### 2.3. Recovery Performance Calculation Method

Water and heat are simultaneously recovered during the process of gas scouring in a ceramic membrane tube. Water and heat fluxes and the recovery rate are used to assess the performance of a membrane condenser.

The amount of recycled water can be described by
(1)$$J_{w}=\frac{1}{S}(m_{v,\mathrm{in}}-\frac{\Delta m}{\Delta t})$$
where *J_w_* is the amount of recycled water, kg/(m^2^·h), *m_v_*_,in_ is the water vapor content of inlet flue gas, kg/h, Δ*m* is the weight difference of silica-gel desiccant before and after the experiment, kg, Δ*t* is experiment time, *h*, and *S* is effective membrane acreage, m^2^.

Recycled water rate (*η_w_* (%)) is given by
(2)$$\eta_{w}=(1-\frac{\Delta m}{m_{v,\mathrm{in}}\Delta t})\times 100$$

Recovered heat is made up of two items: the heat carried by the water coolant and the enthalpy of recycled water; the calculation equation is written as follows [35]:(3)$$q=\frac{m_{w}c_{p,w}\left(t_{w,\mathrm{out}}-t_{w,\mathrm{in}}\right)}{S}+J_{w}h_{w,\mathrm{out}}$$
where *q* is the recovered heat, kJ/(m^2^·h), *m_w_* is the water coolant mass flow rate, kg/h, *c_p,w_* is the specific heat capacity of water, kJ/(kg·K), *t_w_*_,in_ is the inlet water coolant temperature, °C, *t_w_*_,out_ is the outlet water coolant temperature, °C, and *h_w_*_,out_ is the enthalpy of recycled water at the outlet temperature of water coolant, kJ/kg.

The maximum available recovered heat comes from three parts: convection heat exchange between gas and membrane tube wall, and sensible and latent heat release during steam condensation. The calculation equation is in the following form:(4)$$q_{\max}=\frac{m_{f,\mathrm{in}}c_{p,f}(t_{f,\mathrm{in}}-t_{w,\mathrm{in}})+m_{v,\mathrm{in}}c_{p,w}(t_{f,\mathrm{in}}-t_{w,\mathrm{in}})+m_{v,\mathrm{in}}r}{S}$$
where *q*_max_ is the theoretical maximum recovered heat, kJ/(m^2^·h), *m_f_*_,in_ is the flue gas inlet mass flow rate, kg/h, *c_p,f_* is the flue gas specific heat capacity, kJ/(kg·K), *t_f_*_,in_ is the entrance flue gas temperature, °C, and *r* is the evaporation latent heat, kJ/kg.

Thus, heat recovery rate (*η_h_* (%)) is given by
(5)$$\eta_{h}=\frac{q}{q_{\max}}\times 100$$

Heat exchange in the membrane tube is complicated due to coexisting convection and conduction heat transfer during the processes of flue gas cooling and vapor condensation. The total heat transfer coefficient (THTC) is adopted to assess the heat exchange performance of the membrane condenser. Its calculation method is of the following form:(6)$$k=\frac{q}{\Delta T}$$
where *k* is the THTC, W/(m^2^·K), and Δ*T* is the logarithmic mean temperature difference (LMTD). According to Yang et al. [36], LMTD can be given by
(7)$$\Delta T=\frac{\left(t_{f,\mathrm{in}}-t_{w,\mathrm{out}}\right)-\left(t_{f,\mathrm{out}}-t_{w,\mathrm{in}}\right)}{In\left(\frac{t_{f,\mathrm{in}}-t_{w,\mathrm{out}}}{t_{f,\mathrm{out}}-t_{w,\mathrm{in}}}\right)}$$
where *t_w_*_,out_ is the water coolant outlet temperature, °C, and *t_f_*_,out_ is the flue gas exit temperature, °C.

### 2.4. Uncertainty Analysis

The testing uncertainty may cause experimental errors. Uncertainty analysis was performed in order to preserve the accuracy of experimental results in the study. The direct testing parameters included nitrogen flow rate $Q_{N_{2}}$, relative humidity *φ*, silica-gel desiccant weight before experiment *m*_1_, silica-gel desiccant weight after experiment *m*_2_, water flow rate $Q_{w}$, $t_{w,\mathrm{in}}$, $t_{w,\mathrm{out}}$, $t_{f,\mathrm{in}}$, and $t_{f,\mathrm{out}}$.

The relative uncertainty of the amount of recycled water $\Delta J_{w}$ can be determined by
(8)$$\Delta J_{w}=\frac{\sqrt{{\left(\frac{\partial J_{w}}{\partial Q_{N_{2}}}\Delta Q_{N_{2}}\right)}^{2}+{\left(\frac{\partial J_{w}}{\partial \varphi }\Delta \varphi \right)}^{2}+{\left(\frac{\partial J_{w}}{\partial m_{2}}\Delta m_{2}\right)}^{2}+{\left(\frac{\partial J_{w}}{\partial m_{1}}\Delta m_{1}\right)}^{2}}}{J_{w}}$$

The relative uncertainty of recovered heat Δq can be determined by
(9)$$\Delta q=\frac{\sqrt{\begin{array}{l} {\left(\frac{\partial q}{\partial Q_{w}}\Delta Q_{w}\right)}^{2}+{\left(\frac{\partial q}{\partial t_{w,\mathrm{in}}}\Delta t_{w,\mathrm{in}}\right)}^{2}+{\left(\frac{\partial q}{\partial t_{w,\mathrm{out}}}\Delta t_{w,\mathrm{out}}\right)}^{2}\\ +{\left(\frac{\partial q}{\partial Q_{N_{2}}}\Delta Q_{N_{2}}\right)}^{2}+{\left(\frac{\partial q}{\partial \varphi }\Delta \varphi \right)}^{2}+{\left(\frac{\partial q}{\partial m_{1}}\Delta m_{1}\right)}^{2}+{\left(\frac{\partial q}{\partial m_{2}}\Delta m_{2}\right)}^{2} \end{array}}}{q}$$

The relative uncertainty of the maximum available recovered heat $q_{\max}$ can be determined as follows:(10)$$\Delta q_{\max}=\frac{\sqrt{\begin{array}{l} {\left(\frac{\partial q_{\max}}{\partial Q_{N_{2}}}\Delta Q_{N_{2}}\right)}^{2}+{\left(\frac{\partial q_{\max}}{\partial t_{w,\mathrm{in}}}\Delta t_{w,\mathrm{in}}\right)}^{2}+{\left(\frac{\partial q_{\max}}{\partial t_{f,\mathrm{in}}}\Delta t_{f,\mathrm{in}}\right)}^{2}\\ +{\left(\frac{\partial q_{\max}}{\partial Q_{N_{2}}}\Delta Q_{N_{2}}\right)}^{2}+{\left(\frac{\partial q_{\max}}{\partial \varphi }\Delta \varphi \right)}^{2}+{\left(\frac{\partial q_{\max}}{\partial m_{1}}\Delta m_{1}\right)}^{2}+{\left(\frac{\partial q_{\max}}{\partial m_{2}}\Delta m_{2}\right)}^{2} \end{array}}}{q_{\max}}$$

Through calculation, the maximum relative uncertainties of the amounts of recycled water, recovered heat, and the maximum available recovered heat were 3.31%, 7.87%, and 1.12%, respectively.

## 3. Results and Discussion

### 3.1. Flue Gas Flow Rate

Figure 3a shows that, when the flue gas increased from 6.25 × 10^−5^ to 3.125 × 10^−4^ kg/s, the amount of recycled water increased linearly from 0.55 to 2.57 kg/(m^2^·h). Conversely, the recycled water rate dropped from 83.0% to 75.2%. Under the same flue gas temperature and flue gas relative humidity conditions, by increasing the gas flow rate, the water vapor content carried by the flue gas increased. Hence, as the amount of flue gas entering the cavity body of membrane module increased, so did the water recovery flux. However, the recycled water rate changed in the opposite direction, and it dropped upon increasing the flue gas flow rate. The reason is that a larger gas flow rate results in the gas spending less time in the module. A part of water vapor is not condensed and then discharges out of the membrane module. The gas humidity ratio decreased with the increase in gas flow rate [30]. The moisture content of the gas did not increase proportionally with the increase of gas flow rate; as a result, water flux decreased slightly, which was the main factor leading to the water recovery flux variation showing an opposite trend to that of this study.

In the experimental conditions, the recovered heat and its recovery rate (Figure 3b) were consistent with the trends of the amount of recycled water and recycled water rate, respectively. Increasing the flue gas flow rate meant that more heat flowed into the membrane module body. In addition, heat transfer was strongly linked with the amount of recovered water; in other words, more water recovered led to more heat being recovered. Therefore, the recovered heat increased upon growth of the gas flow rate. However, the recovered heat rate exhibited a decreasing trend due to a reduction in flue gas residence time.

As seen in the Figure 3c, when the flue gas flow rate increased from 6.25 × 10^−5^ to 3.125 × 10^−4^ kg/s, the total heat transfer coefficient (THTC) increased from 82.2 to 134.5 W/(m^2^·K). As mentioned above, THTC was used to evaluate heat exchange performance of membrane tube. The recovered heat is related to the total heat transfer coefficient and heat transfer temperature difference. Under the experimental conditions of this section, the change in heat transfer temperature difference was slight; thus, the changing trend of THTC was consistent with that of recovered heat.

### 3.2. Flue Gas Temperature

As can be seen in Figure 4a, both the amount of recycled water and the recycled water rate increased significantly as flue gas temperature increased. Within the temperature range of the experimental study, the amount of recycled water increased significantly from 0.80 to 2.93 kg/(m^2^·h). Meanwhile, the recycled rate increased from 67.1% to 81.9%. Water vapor content in wet saturated gas augmented significantly with growth of temperature [4]. A higher temperature resulted in more water vapor content. Furthermore, an increase in gas temperature meant that the temperature difference between the flue gas and tube wall increased; thus, the driving force of transportation increased, prompting more water to be recovered.

As the temperature increased from 40 to 60 °C, heat flux went from 1.44 to 3.63 kW/m^2^; by contrast, heat recovery rate went from 80.3% to 68.5% (Figure 4b). A higher flue gas temperature denoted greater enthalpy of the flue gas, as well as more sensible heat to be released. Accompanied by the amount of recycled water increasing noticeably, the increase in recovered heat was obvious. As the temperature increased from 40 to 50 °C, the recovered heat increased by 50.8%. When it increased to 60 °C, heat flux increased by 67.1%. The maximum recoverable heat was almost entirely composed of latent heat released during the vapor condensation process. For saturated gas, the water vapor content carried by the same gas flow rate was only related to the partial pressure of water vapor, and the partial pressure of water vapor increased exponentially with the increase in gas temperature; thus, the water vapor content carried in the flue gas varied parabolically with flue gas temperature, as shown in Figure 5. Therefore, with the increase in flue gas temperature, the water vapor content increased exponentially, and the heat released during the condensation process also increased exponentially. The growth trend of recovered heat was slower than that of the maximum recoverable heat; therefore, the heat recovery rate was reduced.

In the experimental conditions, Figure 4c indicates that the THTC increased from 110.9 to 141.6 W/(m^2^·K). Compared to Figure 4b,c, it can be found that the trend of the THTC was consistent with the recovered heat. The increment of THTC in the latter stage was significantly greater than that in the former stage. This is because increasing the heat exchange temperature difference could improve the efficiency of the heat exchanger. A greater temperature difference led to a higher heat transfer efficiency.

The exhaust gas temperature after wet desulfurization is generally about 50 °C [37]. In order to provide directions for practical application, the study hereby investigated the temperature from 40 to 60 °C.

### 3.3. Water Coolant Flux

Figure 6 depicts the influence of water coolant flux on recovering moisture and heat. When the water coolant flux increased from 8.32 × 10^−3^ to 3.327 × 10^−2^ kg/s, as shown in Figure 6a, the amount of recycled water increased from 1.50 to 1.66 kg/(m^2^·h). The recycled water flux changed little with the variation of water coolant flux, especially when water coolant flux was more than 1.664 × 10^−2^ kg/s. When the water coolant flux was 1.664 × 10^−2^ kg/s, its mass flow rate was 89 times that of the flue gas, which far exceeded flue gas mass flow. The effect of increasing the water coolant flux on the amount of recycled water was gradually weakened. Similar results were reported by Chen et al. [4]. The increasing trend of recycled water rate was consistent with that of the amount of recycled water. Under the experimental conditions, the recycled water rate increased from 73.3% to 82.8%.

As the water coolant flux increased, the flow velocity in the tube increased, the heat transfer coefficient of the inner surface of the membrane tube increased, the heat transfer capacity of the membrane tube improved, and the recovered heat increased. Because the flue gas flow rate was small, the flue gas enthalpy value was correspondingly low, such that the increase in the amount of heat recovery was small, which was also the reason that the amount of recycled water did not change much. As presented in Figure 6b, the recovered heat was composed of the enthalpy increase of water coolant and the enthalpy value of the recycled water. Since the amount of recycled water was small and did not change much, the proportion of the enthalpy of the amount of recycled water in the recovered heat was very small. The recovered heat was principally composed of the heat absorbed by the water coolant, which accounted for 97.1–98.4%. In the experimental conditions, with an increase in water coolant flux, the THTC increased from 98.0 to 159.7 W/(m^2^·K) (Figure 6c).

### 3.4. Water Coolant Temperature

Figure 7a describes that when water coolant temperature increased, the amount of recycled water and the recycled water rate gradually decreased. The graph can be divided into two different downward trends. When the water coolant temperature increased from 15 to 30 °C, the amount of recycled water declined from 1.63 to 1.55 kg/(m^2^·h), decreasing by 5%. However, when the temperature reached 35 °C, the amount of recycled water was 1.41 kg/(m^2^·h), decreasing dramatically by 9%. For heat exchangers, a higher temperature of the cooling medium leads to a worse cooling capacity. When water coolant temperature was higher than 30 °C, the heat transfer temperature difference between flue gas and water gradually decreased, and the cooling effect deteriorated, leading to a sharp decline in water vapor condensation rate, as the amount of recycled water dropped sharply. This indicated that increasing water coolant temperature led to a decrease in the condensation rate of water vapor. A higher temperature resulted in a more serious condensation deterioration phenomenon.

The recycled water rate had the same trend as that of the amount of recycled water when water coolant temperature varied (Figure 7b). Under the same conditions of flue gas, the water content was equal, and the decrease in the amount of recycled water resulted in a decrease in recycled water rate. The recovered heat and heat recovery rate were almost linearly reduced with the water coolant temperature increasing from 15 to 30 °C, unlike the tendency of the amount of recycled water, which decreased at high water coolant temperatures. As mentioned in Section 3.3, the heat absorbed by the water coolant played the main role in overall heat recovery. Furthermore, the heat absorbed by water coolant decreased linearly as water coolant temperature increased, as seen in Figure 7b. Thus, the recovered heat decreased linearly upon an increase in water temperature.

Figure 7c shows that the THTC increased from 107.6 to 224.3 W/(m^2^·K) with the water coolant temperature rising from 15 to 35 °C. This was mainly caused by the different decreasing trends of the recovered heat and the logarithmic mean temperature difference. With the increase in water temperature from 15 to 35 °C, the recovered heat dropped from 8.65 to 5.30 kW/m^2^ (by about 38.7%). On the other hand, the logarithmic mean temperature difference decreased from 22.3 to 6.6 (by about 70.6%). Therefore, the total heat transfer coefficient tended to increase. The result is opposite to that in reference, in which the rise of the outlet flue gas temperature of the membrane module was small, and the temperature difference of the heat transfer was also small [35]. Hence, the total heat transfer coefficient was gradually reduced. In addition, the results are probably related to the flow direction of the flue gas and water. In this study, the flue gas flowed parallel and counter-currently to the water coolant, while, the flue gas vertically flushed the membrane tube, i.e., the flue gas is perpendicular to the direction of water coolant flow [35].

### 3.5. Comparison of Different Research Results

The water recovery performance obtained in this study is compared with other research results in Table 3. The pore sizes of the ceramic membranes and experimental conditions in each study were different. A nanoporous ceramic membrane tube, which was an inner coating membrane, was used [4,21,30]. Gao et al. [35] used the same membrane tube as this article, which was an outer coating membrane with an average pore size of 1 μm. However, a gas-fired boiler flue gas was adopted [35]. The flue gas flow rate/membrane area ratio was much larger than that in this article. Therefore, the amount of recycled water was greater than the results of this work.

The recovery performance of different research results varied with operational conditions as shown in Table 4. The change in recovery performance with operating conditions in this study was almost consistent with other studies; however, several trends were inconsistent. For instance, the THTC increased with the increase in coolant water temperature in this paper. The main reason was that the variation trend of the logarithmic mean temperature difference was greater than that of recovered heat.

## 4. Conclusions

In this paper, the moisture and waste heat recovery performance of a 1-μm-pore ceramic membrane tube were studied. The influencing factors including flue gas flow rate, flue gas temperature, water coolant flux, and water coolant temperature on the recovery performance of the membrane module were investigated. The following conclusions were drawn:With the flue gas flow rate increasing, the amount of recycled water and recovered heat increased linearly, while the recycled water rate and heat recovery rate dropped.The amount of recycled water, recycled water rate, and recovered heat increased with the increase in flue gas temperature. The growth trend of recovered heat was slower than that of the maximum recoverable heat, which resulted in a decrease in heat recovery rate.Along with water coolant temperature growth, the amount of recycled water, recycled water rate, recovered heat, and heat recovery rate decreased. A higher temperature resulted in a more serious deterioration of water vapor condensation. When the water coolant temperature exceeded 30 °C, the amount of recycled water dropped sharply.Under the experimental conditions, the maximum amounts of recycled water, recovered heat, and total heat transfer coefficient were 2.93 kg/(m^2^·h), 3.63 kW/m^2^, and 224.3 W/(m^2^·K), respectively.

## Figures and Tables

**Figure 1 materials-13-00804-f001:**
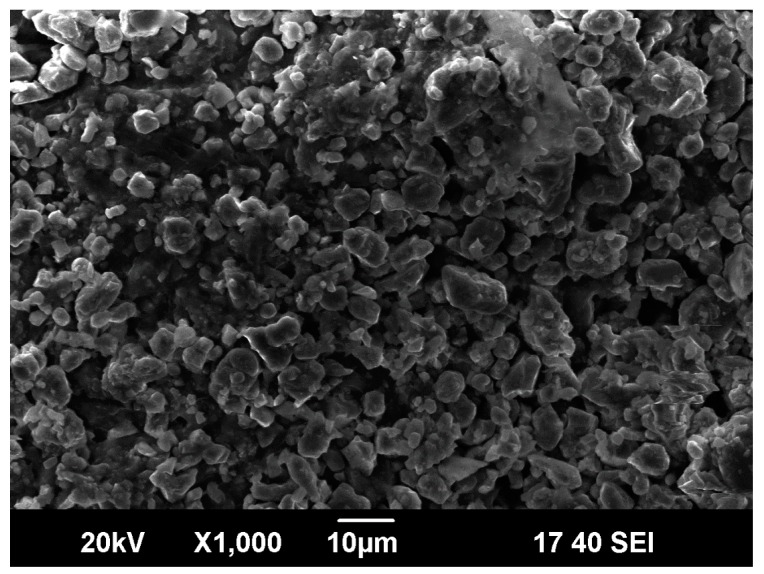
Microstructure of 1-μm-pore ceramic membrane.

**Figure 2 materials-13-00804-f002:**
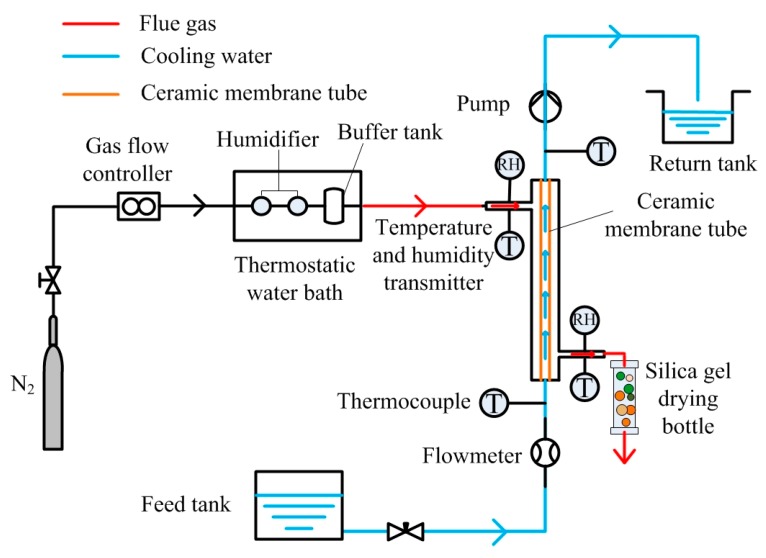
Schematic of experimental system.

**Figure 3 materials-13-00804-f003:**
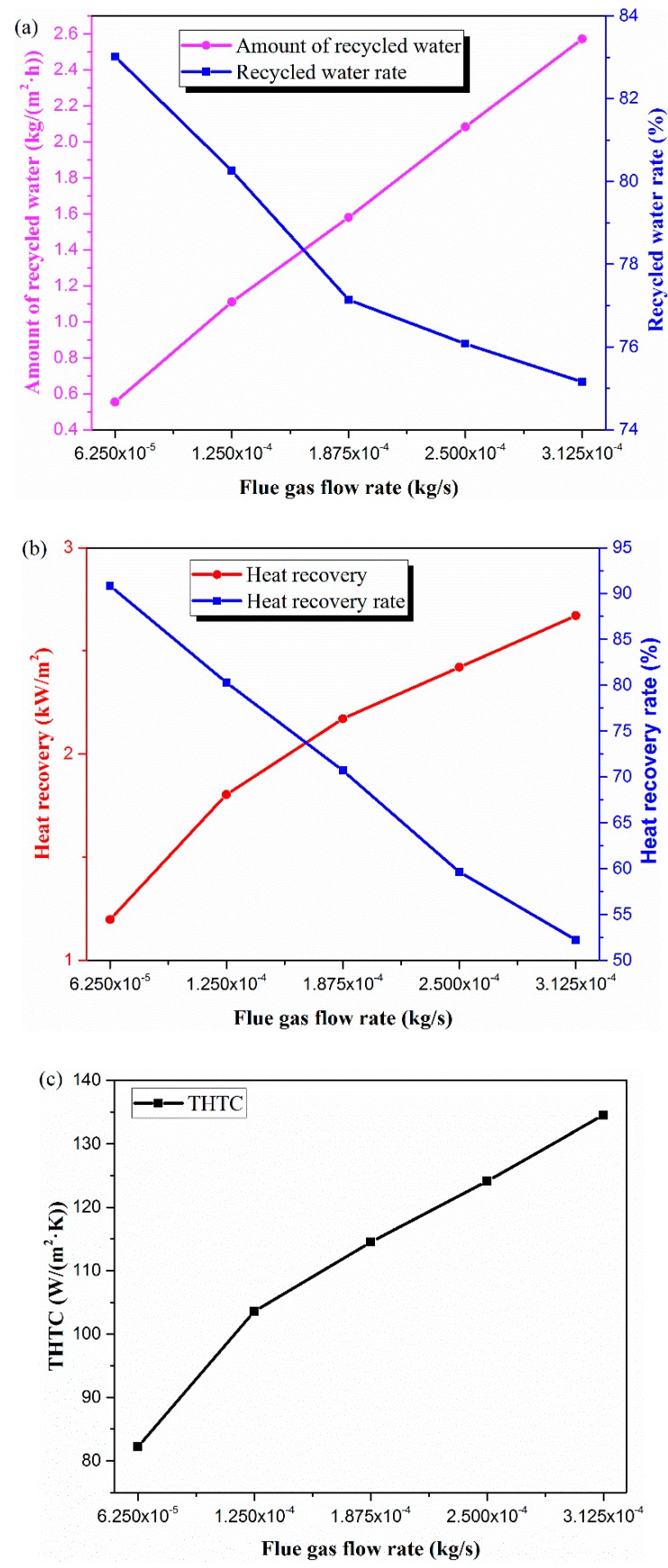
Effect of flue gas flow rate on (**a**) water, (**b**) heat, and (**c**) total heat transfer coefficient (THTC). (Experimental conditions: flue gas temperature 50 °C, water coolant flux 1.664 × 10^−2^ kg/s, water coolant temperature 20 °C).

**Figure 4 materials-13-00804-f004:**
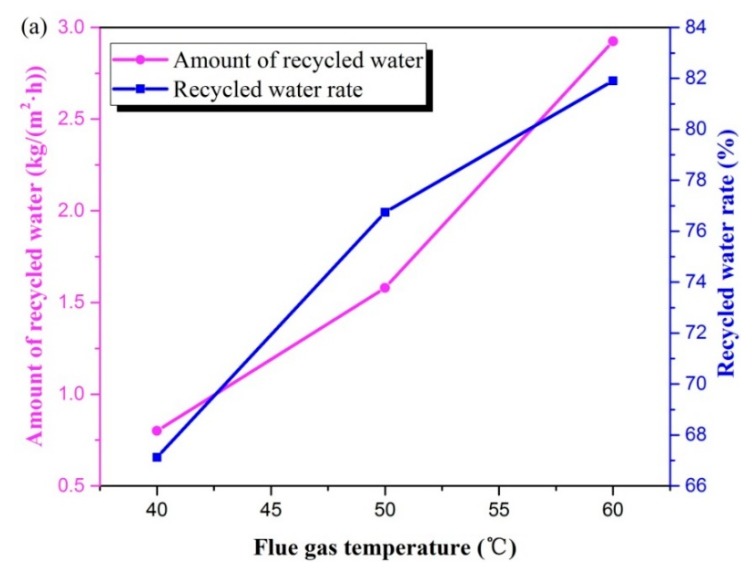
Variables changes with flue gas temperature in terms of (**a**) water, (**b**) heat, and (**c**) THTC. (Experimental conditions: flue gas flow rate 1.875 × 10^−4^ kg/s, water coolant flux 1.664 × 10^−2^ kg/s, water coolant temperature 20 °C).

**Figure 5 materials-13-00804-f005:**
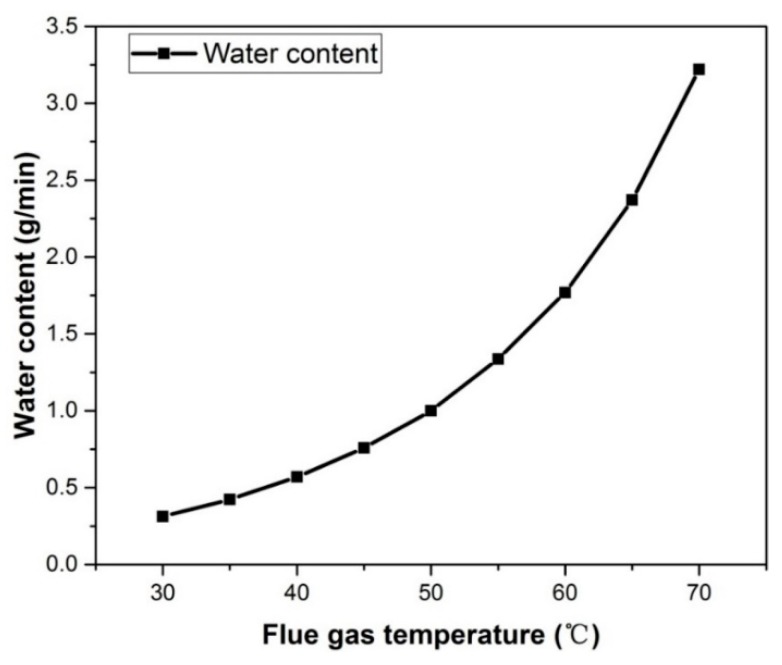
Vapor content change with flue gas temperature (Flue gas flow rate 1.875 × 10^−4^ kg/s).

**Figure 6 materials-13-00804-f006:**
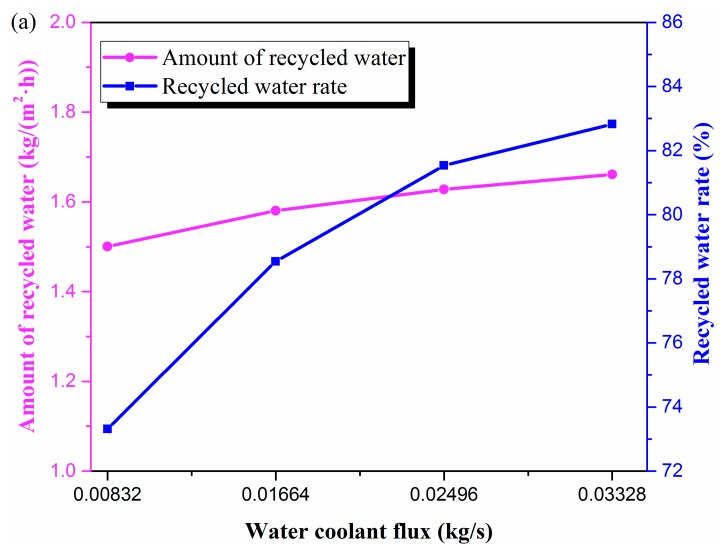
Membrane performance changes with water coolant flux in terms of (**a**) water, (**b**) heat, and (**c**) THTC. (Experimental conditions: flue gas flow rate 1.875 × 10^−4^ kg/s, flue gas temperature 50 °C, water coolant temperature 20 °C).

**Figure 7 materials-13-00804-f007:**
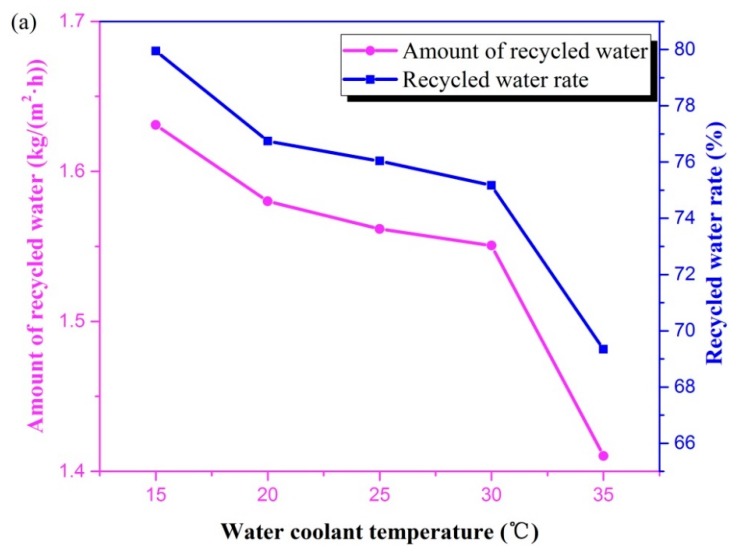
Effect of water coolant temperature in terms of (**a**) water, (**b**) heat, and (**c**) *THTC*. (Experimental conditions: flue gas flow rate 1.875 × 10^−4^ kg/s, flue gas temperature 50 °C, water coolant flux 1.664 × 10^−2^ kg/s).

**Table 1 materials-13-00804-t001:** Experimental operational parameters.

Item	Unit	Value
Flue gas flow rate	kg/s	6.25 × 10^−5^ to 3.125 × 10^−4^
Flue gas temperature	°C	40; 50; 60
Relative humidity	%	100
Water coolant flux	kg/s	8.32 × 10^−3^ to 3.327 × 10^−2^
Water coolant temperature	°C	15–35

**Table 2 materials-13-00804-t002:** Parameters of experimental apparatus.

Experimental Apparatus	Model	Parameters	Precision	Manufacturer
Gas flow controller	D07-9E	30 SLM; Max pressure:3 MPa	±2%	Beijing Sevenstar, Beijing, China
Electric thermostatic water tank	HH.W21.600	Rated power: 750 W ± 10%;	±0.5 °C	Shanghai shuli, Shanghai, China
Temperature and humidity transmitter	TH-21E	Temperature range: −40 to 125 °C Relative humidity range: 0–100%	≤±0.2 °C ≤±2%	Guangzhou Anymetre, Guangzhou, China
Eight-loop digital display device	HT-MK807-01-23-KL	-	0.5% FS	Hantang Precision Instrument, Wuxi, China
Thermocouple	PT100	−50 to 200 °C	A Class	Hangzhou Sinomeasure, Hangzhou, China
Miniature electric diaphragm pump	PLD-1205	Maximum flow rate: 3.2 L/min	-	Shijiazhuang Pulandi, Shijiazhuang, China
Flowmeter	LZT-M15	Range: 0.2–2.0 L/min	≤±4%	Vakada, Suzhou, China

**Table 3 materials-13-00804-t003:** Different research results under different experimental conditions.

Reference	Pore Size	Membrane Area (m^2^)	Coating	Component	Water Flux kg/(m^2^·h)	Experimental Conditions
[4]	20 nm	0.025	Inner coating	N_2_/water vapor	5.7	Inlet gas temperature and flow rate were 60 °C and 14 L/min, respectively; coolant water temperature and flow rate were 16 °C and 2 L/min, respectively
[21]	6–8 nm	0.0021	Inner coating	Air/water vapor	15.8	Inlet gas temperature and flow rate were 75 °C and 4 L/min, respectively; coolant water temperature and flow rate were 33 °C and 5 L/h, respectively
[30]	7 nm	0.0021	Inner coating	Air/water vapor	4.5	Inlet gas temperature and flow rate were 100 °C and 6.7 L/min, respectively; coolant water flow rate was 3.3 L/h
[35]	1 μm	0.7	Outer coating	Gas-fired boiler flue gas	15.8	Inlet gas temperature and flow rate were 46 °C and 1600 m^3^/h, respectively; coolant water temperature and flow rate were 23 °C and 1150 L/h, respectively
This paper	1 μm	0.0294	Outer coating	N_2_/water vapor	2.6	Inlet gas temperature and flow rate were 50 °C and 15 L/min, respectively; coolant water temperature and flow rate were 20 °C and 1 L/min, respectively

**Table 4 materials-13-00804-t004:** Recovery performance variation with operational conditions.

Ref		Flue gas Flow Rate	Flue Gas Temperature	Coolant Water Flow Rate	Coolant Water Temperature
[4]	Water flux	Increased linearly	Increased exponentially	Changed little	Decreased parabolically
	Heat flux	Increased linearly	Increased exponentially	Increased	-
	THTC	-	-	-	-
[21]	Water flux	Increased linearly	Increased exponentially	Increased slightly	-
	Heat flux	Increased exponentially	Increased exponentially	increased linearly	-
	THTC	-	-	-	-
[30]	Water flux	Decreased linearly	Increased linearly	Increased linearly	Decreased linearly
	Heat flux	Decreased linearly	Increased linearly	Increased linearly	Decreased linearly
	THTC	Decreased parabolically	Decreased lightly	Increased linearly	Decreased linearly
[35]	Water flux	Increased linearly	Increased linearly	Increased linearly	Decreased linearly
	Heat flux	Increased linearly	Increased linearly	Increased linearly	Decreased linearly
	THTC	Increased linearly	Changed little	Increased linearly	Decreased linearly
This paper	Water flux	Increased linearly	Increased exponentially	Increased lightly	Decreased lightly
	Heat flux	Increased linearly	Increased exponentially	Increased linearly	Decreased linearly
	THTC	Increased linearly	Increased exponentially	Increased linearly	Increased

Increased/decreased linearly means that the recovery performance increased/decreased linearly with operational conditions, by more than 20%. Increased/decreased lightly means that the recovery performance increased/decreased with operational conditions, by less than 10%. Increased/decreased exponentially/parabolically means that the recovery performance increased/decreased exponentially/parabolically with operational conditions, by more than 20%.

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
