# Peer review of "Experimental Study on Water Recovery from Flue Gas Using Macroporous Ceramic Membrane"

_materials, 2020, doi:10.3390/ma13030804_

Round 1
Reviewer 1 Report
Dear Author(s),
I ask you to make some improvements and some corrections of some mistakes and details as outlined below.
In general: Please check the use of articles a (an) and the. English is not my first language so I am not qualified to correct it. However, my recommendation is in the attached file. I also recommend that the manuscript be corrected and linguistically improved by a native English speaker.
In general: Please mention the relation to the Figure(s) in the manuscript. For example, in line 108: The microstructure of the membrane tube presents the typical porous structure (Figure 1).
or: Figure 1 presents ……
In general: Please use the capital first letter in Figure captions. For example, in line 112: Figure 1. Microstructure of 1 μm pore size ceramic membrane.
Line 3: Do you mean microporous or macroporous?
Line 92: Do you mean microporous or macroporous?
Line 107: Missing reference
Line 107: The microstructure (Figure 1) of the membrane tube presents the typical porous structure.
Line 112: Figure 1. Microstructure of 1 μm pore size ceramic membrane.
Line 124: Missing reference
Line 139: kPa.
Line 194: (8)
Line 196: (9)
Line 199: (10)
Line 200: Do you mean Through or Though?
Line 205: Missing reference
Line 251: Missing reference
Line 266: Missing reference
Line 272: Missing reference
Line 289: Missing reference
Line 303: Missing reference
Line 315: Missing reference
Line 333: Missing reference

Author Response
Response to Reviewer 1 Comments
I ask you to make some improvements and some corrections of some mistakes and details as outlined below.
Point 1: In general: Please check the use of articles a (an) and the. English is not my first language so I am not qualified to correct it. However, my recommendation is in the attached file. I also recommend that the manuscript be corrected and linguistically improved by a native English speaker.
Response 1: Really thank you for your comments and recommendation. Based on your recommendation, We checked and corrected the use of articles a (an) and the in the manuscript. In addition, We invited a professional to assist us in checking and correcting the manuscript. Partial correction results are as follows:
In this work, the ceramic membrane tube with a pore size of 1 μm was used to conduct experimental research on moisture and afterheat recycle from flue gas. For example, in the southern region, the cooling system has been changed from open type to closed type. And in the northern region, the cooling system was changed from closed type to air cooling type. The boiler flue gas of thermal power plants usually contains a large amount of vapor and lots of latent heat. Besides, the treatment of precipitate formed by the contact between the flue gas and the absorbent is technically difficult. Brunetti et al. [15] experimentally examined the influence of ΔT (temperature difference between the flue gas and the module) on the water recovery rate. Chen et al. [22] experimentally studied the water recovery performance of ceramic membranes with pore sizes with pore sizes of 20, 30, 50, and 100 nm. It was found that the recovery performance of the 20 nm pore-sized membrane was better. The sintering temperature is high, the manufacturing process is complex, causing a high product cost. In this paper, a single membrane tube with a pore size of 1 μm was used for the experimental study of extracting moisture and waste heat from the exhaust gas. The results of this study can provide guidance for recovering water and heat from power plant exhaust by utilizing ceramic membrane. However, the recycled water rate is oppositely changing, and drops with the increasing of the flue gas flow rate. Increasing the flue gas flow rate flow rate means that more heat flows into the membrane module body.
Point 2: In general: Please mention the relation to the Figure(s) in the manuscript. For example, in line 108: The microstructure of the membrane tube presents the typical porous structure (Figure 1). or: Figure 1 presents …….
Response 2: Really thank you for your comments that are of great help to the manuscript. We knew about the importance of relationship between Figure(s) and content. Therefore, we added the corresponding description, for example:
After revised:
Figure 1 presents the scanning electron microscopy (SEM) graph of 1 μm average pore size ceramic membrane. The microstructure (Figure 1) of the membrane tube presents a typical porous structure. Figure 3(a) shows that when the flue gas rises from 6.25×10-5 to 3.125×10-4 kg/s…the heat recovery and its recovery rate (Figure 3(b)) are consistent with the trends of the amount of recycled water and recycled water rate, respectively. As seen in the Figure 3(c), when the flue gas flow rate increases from 6.25×10-5 to 3.125×10-4 kg/s. As can be seen in Figure 4(a), both the amount of recycled water and recycled water rate increase significantly as flue gas temperature rises… As the temperature is increased from 40 to 60 °C, heat flux goes up from 1.44 to 3.63 kW/m2, by contrast, heat recovery rate goes down from 80.3 to 68.5% (Figure 4(b))…In the experimental conditions, Figure 4(c) indicates that the THTC increases from 110.9 to 141.6 W/(m2K). Figure 6 depicts the influence of water coolant flux on recovering moisture and heat…When the water coolant flux increases from 8.32×10-3 to 3.327×10-2 kg/s, as shown in Figure 6(a)…As presented in Figure 6(b)…the THTC is increased from 98.0 to 159.7 W/(m2K) (Figure 6(c)). The recycled water rate has the same variation trend as that of the amount of recycled water when water coolant temperature varies (Figure 7(b)). Figure 7(c) shows that the THTC is increased from 107.6 to 224.3 W/(m2K) with the water coolant temperature rising from 15 to 35°C.
Point 3: In general: Please use the capital first letter in Figure captions. For example, in line 112: Figure 1. Microstructure of 1 μm pore size ceramic membrane.
Response 3: The reviewer’s comment has been accepted by correcting the first letter in Figure captions to capital in the whole manuscript.
After revised:
Figure 1. Macrostructure of 1 μm pore size ceramic membrane. Figure 2. Schematic of experimental system. Figure 3. Effect of flue gas flow rate. Figure 4. Variables changes with flue gas temperature. Figure 5. Vapor content varies with flue gas temperature Figure 6. Membrane performance changes with water coolant flux. Figure 7. Effect of water coolant temperature.
Point 4: Line 3: Do you mean microporous or macroporous?
Line 92: Do you mean microporous or macroporous?
Response 4: The reviewer’s comment has been accepted with explanations as follows:
We refer to a literature (Cychosz K A , Guillet-Nicolas, Rémy, García-Martínez, Javier, et al. Recent advances in the textural characterization of hierarchically structured nanoporous materials[J]. Chem. Soc. Rev. 2016.), where the author said based on the IUPAC recommendations from 2015, pores are divided into three groups according to their pore width: micropores of widths less than 2 nm, mesopores of widths between 2 and 50 nm and macropores of widths greater than 50 nm. In the manuscript, the average pore diameter of the used ceramic membrane is 1μm, so it is called macroporous membrane.
Point 5:
Line 107: The microstructure (Figure 1) of the membrane tube presents the typical porous structure.
Line 112: Figure 1. Microstructure of 1 μm pore size ceramic membrane.
Response 5: Really thank you for your Comment. In the original manuscript, we used the "structure" to indicate the structure of the membrane tube. Figure 1 is the result of observation with a scanning electron microscopy at a magnification of 1000 times. The length of the white line in the figure represents 10 μm, so the term "microstructure " is more appropriate. The original Line 107: Missing reference is "Figure 1", here, Figure 1 was also added after the microstructure. Therefore, the original Line 112: graph of 1 μm pore size ceramic membrane was corresponding changed to "Figure 1. Microstructure of 1 μm pore size ceramic membrane.".
After revised:
Figure 1 presents the scanning electron microscopy (SEM) graph of 1 μm average pore size ceramic membrane. The microstructure (Figure 1) of the membrane tube presents a typical porous structure.
Figure 1. Microstructure Microstructure of 1 μm pore size ceramic membrane.
Point 6:
Line 107: Missing reference
Line 124: Missing reference
Line 205: Missing reference
Line 251: Missing reference
Line 266: Missing reference
Line 272: Missing reference
Line 289: Missing reference
Line 303: Missing reference
Line 315: Missing reference
Line 333: Missing reference
Response 6: Really thank you for your Comment. In the original manuscript, we used the "Cross-reference " function in Microsoft Word. It may be that the references of Figure were lost during submission. We added the missing references in the revised version.
Figure 1 presents the scanning electron microscopy (SEM) graph of 1 μm average pore size ceramic membrane. Figure 2 illustrates the experimental system schematic. Figure 3(a) shows that when the flue gas rises from 6.25×10-5 to 3.125×10-4 kg/s. As can be seen in Figure 4(a), both the amount of recycled water and recycled water rate increase significantly as flue gas temperature rises. thus the water vapor content carried in the flue gas varies parabolically with flue gas temperature, as shown in Figure 5. Compared Figure 4(c) and (b), it is found that the trend of the THTC is consistent with the recovered heat. Figure 6 depicts the influence of water coolant flux on recovering moisture and heat. As presented in Figure 6(b), the recovered heat is composed of enthalpy increase of water coolant and the enthalpy value of the recycled water. Figure 7(a) describes that when water coolant temperature increases, the amount of recycled water and recycled water rate gradually decrease. The recycled water rate has the same variation trend as that of the amount of recycled water when water coolant temperature varies (Figure 7(b)).
Point 7: Line 139: kPa. Do you mean Through or Though?
Response 7: Really thank you for your comments that are of great help to the manuscript. The reviewer’s comment has been accepted by revising the unit and word (correct "kpa" to "kPa " and correct "Though" to "Through ") together with others in the whole manuscript.
Point 8:
Line 194: (8)
Line 196: (9)
Line 199: (10)
Response 8: Really thank you for your Comment, which helps us improve the quality of our manuscript. We added the equation number in the revised version.
After revised:
(8)
(9)
(10)

Reviewer 2 Report
The authors have carried out research to broaden the knowledge about the possibility of using ceramic membranes to recovery water and heat from flue gas, with a potentially high application value. Before publication, I propose to introduce the following changes:
In the Abstract:- the authors should take into account the information on testing the membrane with the outer-coating membrane tube and the direction of exhaust gases and cooling water flow;
- the authors provide numerical values for maximum water and heat recovery, indicating the optimal variant for the separation process.
Throughout the entire article, please insert references to the discussed Figures: e.g. Fig. 2 (line 124); Fig. 3a (line 205); Fig. 3b (line 218); and other Please discuss Figure 5. The authors should also harmonize the scale of the x-axis in Figure 6 with the data discussed in section 3.3.Author Response
Response to Reviewer 2 Comments
The authors have carried out research to broaden the knowledge about the possibility of using ceramic membranes to recovery water and heat from flue gas, with a potentially high application value. Before publication, I propose to introduce the following changes:
Point 1: In the Abstract: - the authors should take into account the information on testing the membrane with the outer-coating membrane tube and the direction of exhaust gases and cooling water flowï¼›
-the authors provide numerical values for maximum water and heat recovery, indicating the optimal variant for the separation process.
Response 1: Really thank you for your Comment, which helps us improve the quality of our manuscript. We highlighted the information and numerical values of this work in the Abstract in the revised manuscript.
Here shows the Abstract before and after revised.
Before revised:
Abstract: In this work, the ceramic membrane tube with a pore size of 1 μm was used to conduct experimental research on moisture and afterheat recycle from flue gas. In experiments, the flue gas was artificially prepared. The effects of flue gas flow rate, flue gas temperature, water coolant flux and water coolant temperature on the membrane recovery performance were analyzed. The results indicated that increasing the flue gas flow rate and its temperature, both recycled water and recovered heat were increased. Under different water coolant flux conditions, the amount of recycled water did not change much. The amount of recycled water, recycled water rate, heat recovery and heat recovery rate all decreased as the water coolant temperature was increasing. When the water coolant temperature exceeded 30°C, the amount of recycled water dropped sharply. In the experiments, the maximum total heat transfer coefficient was 224.3 W/(m2•K).
After revised:
Abstract: In this work, the ceramic membrane tube with a pore size of 1 μm was used to conduct experimental research on moisture and waste heat recovery from flue gas. The length, inner/outer diameter, and porosity are 800 mm, 8/12 mm, and 27.2%, respectively. In experiments, the flue gas, which was artificially prepared, flowed on the shell side of membrane module. The water coolant passed through inside the membrane counter-currently with the gas. The effects of flue gas flow rate, flue gas temperature, water coolant flux and water coolant temperature on the membrane recovery performance were analyzed. The results indicated that increasing the flue gas flow rate and its temperature, both recycled water and recovered heat were increased. The amount of recycled water, recycled water rate, heat recovery and heat recovery rate all decreased as the water coolant temperature was increasing. When the water coolant temperature exceeded 30°C, the amount of recycled water dropped sharply. In the experiments, the maximum amount of recycled water, recovered heat, and total heat transfer coefficient were 2.93 kg/(m2•h), 3.63 kW/m2, and 224.3 W/(m2•K), respectively.
Point 2: Throughout the entire article, please insert references to the discussed Figures: e.g. Fig. 2 (line 124); Fig. 3a (line 205); Fig. 3b (line 218); and other Please discuss Figure 5. The authors should also harmonize the scale of the x-axis in Figure 6 with the data discussed in section 3.3.
Response 2: Really thank you for your comments that are of great help to the manuscript. In the original manuscript, we used the "Cross-reference" function in Microsoft Word. It may be that the references of Figure were lost during submission, causing "Error! Reference source not found." to appear multiple times. We added the missing references in the revised version.
After revised:
Figure 1 presents the scanning electron microscopy (SEM) graph of 1 μm average pore size ceramic membrane. Figure 2 illustrates the experimental system schematic. Figure 3(a) shows that when the flue gas rises from 6.25×10-5 to 3.125×10-4 kg/s… In the experimental conditions, the heat recovery and its recovery rate (Figure 3(b)) are consistent with the trends of the amount of recycled water and recycled water rate, respectively… As seen in the Figure 3(c), when the flue gas flow rate increases from 6.25×10-5 to 3.125×10-4 kg/s. As can be seen in Figure 4(a)… heat recovery rate goes down from 80.3 to 68.5% (Figure 4(b))... In the experimental conditions, Figure 4(c) indicates that the THTC increases from 110.9 to 141.6 W/(m2K). The water vapor content carried in the flue gas varies parabolically with flue gas temperature, as shown in Figure 5. Figure 6 depicts the influence of water coolant flux on recovering moisture and heat…When the water coolant flux increases from 8.32×10-3 to 3.327×10-2 kg/s, as shown in Figure 6(a)…As presented in Figure 6(b)…the THTC is increased from 98.0 to 159.7 W/(m2K) (Figure 6(c)). Figure 7(a) describes that when water coolant temperature augments,…The recycled water rate has the same variation trend as that of the amount of recycled water when water coolant temperature varies (Figure 7(b))…And the heat absorbed by water coolant decreases linearly with water coolant temperature increases, as seen in Figure 7(b)… Figure 7(c) shows that the THTC is increased from 107.6 to 224.3 W/(m2K) with the water coolant temperature rising from 15 to 35°C.
We added more discussions about Figure 5 as follows:
For saturated gas, the water vapor content carried by the same gas flow rate is only related to the partial pressure of water vapor, and the partial pressure of water vapor increases exponentially with the increase of gas temperature, thus the water vapor content carried in the flue gas varies parabolically with flue gas temperature, as shown in Figure 5.
Point 3: The authors should also harmonize the scale of the x-axis in Figure 6 with the data discussed in section 3.3.
Response 3: In terms of the scale of the x-axis in Figure 6, the reason why we chose the scale in the original manuscript is that we wanted to keep the coordinates as simple, causing the scale of the x-axis in Figure 6 with the data discussed in section 3.3 unharmonious. In order to harmonize the scale of the x-axis in Figure 6 with the data discussed in section 3.3, we adjusted the scale of the x-axis in Figure 6.
Figure 6. Membrane performance changes with water coolant flux. (a) water, (b) heat, and (c) THTC . (Experiment conditions: flue gas flow rate 1.875×10-4 kg/s, flue gas temperature 50 °C, water coolant temperature 20 °C).
